# Healing of Skin Wounds in Rats Using Creams Based on Symphytum Officinale Extract

**DOI:** 10.3390/ijms25063099

**Published:** 2024-03-07

**Authors:** Sorin Marian Mârza, Adela Maria Dăescu, Robert Cristian Purdoiu, Mădălina Dragomir, Mariana Tătaru, Iulia Melega, Andras-Laszlo Nagy, Adrian Gal, Flaviu Tăbăran, Sidonia Bogdan, Mirela Moldovan, Emoke Pall, Camelia Munteanu, Klara Magyari, Ionel Papuc

**Affiliations:** 1Faculty of Veterinary Medicine, University of Agricultural Science and Veterinary Medicine, 400372 Cluj-Napoca, Romania; sorinmarza@yahoo.com (S.M.M.); adela.daescu@usamvcluj.ro (A.M.D.); robert.purdoiu@usamvcluj.ro (R.C.P.); madalina.dragomir@usamvcluj.ro (M.D.); mariana.tataru@usamvcluj.ro (M.T.); iulia.melega@usamvcluj.ro (I.M.); or anagy@rossvet.edu.kn (A.-L.N.); adrian.gal@usamvcluj.ro (A.G.); alexandru.tabaran@usamvcluj.ro (F.T.); sidoniabogdan@gmail.com (S.B.); emoke.pall@usamvcluj.ro (E.P.); ionel.papuc@usamvcluj.ro (I.P.); 2Department of Biomedical Sciences, Ross University School of Veterinary Medicine, Basseterre P.O. Box 334, Saint Kitts and Nevis; 3Faculty of Pharmacy, Iuliu Haţieganu University of Medicine and Pharmacy, 400012 Cluj-Napoca, Romania; mmoldovan@umfcluj.ro; 4Department of Plant Culture, Faculty of Agriculture, University of Agricultural Science and Veterinary Medicine, 400372 Cluj-Napoca, Romania; 5Interdisciplinary Research Institute on Bio-Nano-Sciences, Babeș-Bolyai University, 400271 Cluj-Napoca, Romania

**Keywords:** *Symphytum officinale*, antioxidant activity, cytotoxicity, antimicrobial effects, skin wounds

## Abstract

Rosmarinic acid is a well-known natural antioxidant and anti-inflammatory compound, and it is one of the polyphenolic compounds found in comfrey plants. Comfrey root also contains allantoin, which helps with new skin regeneration. This study aimed to investigate the healing and skin regeneration process of skin wounds in Wistar rats using creams based on comfrey extract and to correlate the results with active compounds in the extract. The obtained results showed that comfrey root is rich in bioactive compounds, including allantoin, salvianolic acid, and rosmarinic acid, which are known for their great free radical scavenging activity, and the high antioxidant activity of the extract may be mainly due to these compounds. The obtained extract has an antimicrobial effect on *Staphylococcus aureus* (1530.76/382.69), *Escherichia coli* (6123.01/6123.01), and *Pseudomonas aeruginosa* (6123.01/6123.01). The macroscopic evaluation and the histological analysis of the skin defects 14 days after the intervention showed faster healing and complete healing in the skin excisions treated with oil-in-water cream with 20% extract of comfrey as the active ingredient.

## 1. Introduction

*Symphytum officinale* (comfrey) is used in folk medicine, especially in the treatment of skin wounds and fractures. This plant belongs to the Borangiaceae family. It is a tall, hairy, grassy plant; the flowers are reddish-purple, sometimes pinkish-white, and are arranged in unipolar cymes, and it grows in damp places, by meadows, at the water’s edge [1]. Comfrey has countless uses in pathological conditions such as sprains, bone injuries, rheumatism, liver problems, gastritis, ulcers, gout, skin injuries, hematomas, and thrombophlebitis. Comfrey tea is recommended for hepatic disturbance. In the sense of gastritis and ulcers, in Brazil, this tea is widely used. In the USA, root extract is used for skin problems. In different countries, comfrey is drunk as a tonic beverage [2]. Comfrey leaves and roots are used in concentrations between 5% and 20% in creams, mainly for healing superficial wounds. Comfrey root contains a wide variety of chemical constituents, such as carbohydrates, polysaccharides, allantoin, tannins, pyrrolizidine alkaloids, triterpenes, and phenolic acids (rosmarinic acid, caffeic acid, chlorogenic acid, and p-hydroxybenzoic and p-coumaric acids), which can affect its application in medicine [1,3].

Phenolic elements originating from herbal plants have garnered a lot of attention due to their effective anti-inflammatory, antitumor, antioxidant, antiviral, and anticancer agents [4,5,6,7,8]. The main phenols are rosmarinic acid (RA) and salvianolic acids (SAs), represented by salvianolic acids I, A, B, and C. Topical or local application of RA to the skin has demonstrated potential to hasten wound healing and reduce the risk of skin cancer in murine models [9]. Moreover, RA contains catechol groups that are identical to dopamine. It has been demonstrated that cream with RA contents will not only have good tissue adhesion but will also have a variety of activities that promote wound healing based on the structure and bioactive activities of RA [10]. SAs are found in LPS-stimulated THP-1-macrophages, and SAs downregulate the protein expression of TLR4, p-p65, and p-IB and dramatically lower the mRNA expression levels of IL-1, IL-6, and TNF [11]. According to Xiao et al., SAs can reduce and limit the production of ROS by controlling the Nrf2/Keap1 pathway [12]. In a mouse distal middle cerebral artery occlusion (dMCAO) model, it was also demonstrated that SA therapy promotes cerebral angiogenesis and increases microvessel density [13]. Cardiovascular disorders are treated with SA B. In vitro, it prevents ischemia [14]. SA B also has other pharmacological effects, such as preventing platelet aggregation and controlling vascular tone [15]. Patients who suffer from systemic disorders, such as angina pectoris and myocardial infarction, are typically prescribed Fufang Danshen tablets or Fufang Danshen Dripping Pills, which contain SA B [16]. SA B has important anti-inflammatory properties and the capacity to block the TGF signaling pathway [17].

Other phenolic compounds in the root extract of comfrey, such as p-hydroxybenzoic, caffeic, chlorogenic, and p-coumaric acids, also have beneficial effects on human skin fibroblasts [18].

Allantoin is another biologically active component of comfrey, which also has antioxidant and antibacterial effects [19,20].

Comfrey polysaccharides are primarily composed of galactose, arabinose, glucose, and galacturonic acid, indicating that comfrey polysaccharides are non-starch polysaccharides that may pass to the end of the intestinal tract of monogastric animals and be fermented by intestinal microflora [21].

Consuming comfrey has been linked to specific instances of hepatotoxic reactions in people, including liver fibrosis, portal hypertension, and veno-occlusive disorders [22]. On the other hand, when used externally, no negative side effects have been documented. Pharmacokinetic investigations have revealed low cutaneous absorption [23]. Consequently, different health organizations do not advise its internal use [24]. However, comfrey is only sold over the counter in the UK when it is prescribed by medical herbalists. Internal comfrey use is restricted in the USA, Canada, and other European nations, including Germany, Denmark, and Austria. Its use should be limited to 4–6 weeks per year, according to the European Commission [23].

Comfrey has been suggested as an alternative treatment for skin disorders [25], and comfrey cream has even been proven to be moderately beneficial in treating osteoarthritis [2]. Although the safety evaluations of this plant’s side effects were primarily conducted in single case reports and based on animal research utilizing high doses, they have not yet been evaluated in human trials. Therefore, its safety has not been evaluated using human epidemiological approaches and biochemical markers [26].

Considering the previously described properties, we aimed to obtain a natural-based cream for wound healing with antioxidant and bactericide properties. The antioxidant properties are aimed to help control wound oxidative stress and thereby accelerate wound healing. Bacterial infections are widely known to be extremely detrimental to wound healing. Superbugs, or germs that are resistant, multiresistant, and panresistant to antimicrobials, are thought to be the biggest issue in recent years. Most of these microorganisms exhibit significant zoonotic potential [27]. Infections with ESKAPE-E, also known as Enterobacter spp., Staphylococcus aureus, Klebsiella pneumoniae, Acinetobacter baumannii, Pseudomonas aeruginosa, Enterobacter spp., and Escherichia coli, are the primary causes of these pathologies [28]. These pathogens have significant intrinsic resistance as well as the potential to acquire resistance. The ability of these microorganisms to bind to tissues and later form a biofilm makes ESKAPE-E infections challenging to avoid and treat [29]. Three-dimensional bacterial communities that develop on both biotic and abiotic surfaces constitute bacterial biofilm [30]. Thus, an extract from the roots of comfrey was obtained, from which the content of polyphenols and allantoin was evaluated, followed by an evaluation of antioxidant and bactericidal properties (Figure 1). The extract was introduced to an oil-in-water disperse system, and its effect on wound healing in Wistar rats was evaluated (Figure 1). Therefore, the novelty of this study consists in its histological evaluation of two stages of the healing and skin regeneration process, using an oil-in-water cream based on comfrey extract in different concentrations, an aspect that allows for establishing the concentration that offers the best results.

## 2. Results and Discussion

### 2.1. Comfrey Extract Structural Evaluation and In Vitro Assays

#### 2.1.1. Determination of Total Polyphenol and Allantoin Content of Comfrey Extract

The determination of the total content of polyphenols using the Folin-Ciocalteu spectrophotometric method indicated a value of 7100 μg/mL in the extract of SyOf.

The UV-Vis and FT-IR spectra revealed the presence of allantoin, SAs, and RA in the SyOf extract (Figure 2). The UV-Vis spectrum (Figure 2A) has two maxima at 236 and 304 nm, with the shoulder at 340 nm. According to the literature [31,32], these absorption bends are specific to SAs and RA. The shoulder at 217 nm can be assigned to the presence of the allantoin [33]. The SyOf extract has a typical FT-IR spectrum of organic compounds, which can be assigned to the presence of allantoin, SAs, and RA. For example, the band at 1635 cm^−1^ can be assigned to the stretching vibration of C=O groups, the bands between 1200 and 900 cm^−1^ can be attributed to the stretching vibration of C-C, and the band at 1273 cm^−1^ to the asymmetric stretching vibration of C-O from SAs and RA. The presence of allantoin is proven by the presence of the shoulder at 1772 cm^−1^ due to stretching vibration of C=O and to the bands at 1171, 1269, and 597 cm^−1^ assigned to C-C stretching, C-N stretching, and C-N bending vibration, respectively [34].

Identification of polyphenolic compounds was performed by LC-ESI^+^-MS analysis, based on maximum absorption, elution order, MS signals (molecular peaks and characteristic fragments), and existing literature data. Six phenolic compounds from the class of hydroxycinnamic acids were identified in the aqueous extract of SyOf (Table 1, Appendix A), of which salvianolic acid I and salvianolic acid C were present in the highest concentrations: 2382.23 ± 115.23 μg/mL and 1641.29 ± 80.06 μg/mL, respectively. Significant amounts were also recorded for RA (1055.02 ± 42.20 μg/mL) and SA B (641.83 ± 51.34 μg/mL). The lowest concentrations were represented by SA A (279.50 ± 27.93 μg/mL) and caffeic acid (123.162 ± 12.40 μg/mL). The total content of phenolic compounds in the extract is 6123.05 μg/mL.

The allantoin content was identified using HPLC analysis, and it obtained a considerable concentration of allantoin (8228.023 ± 41 μg/mL) from the class imidazole (Appendix A).

Thus, the obtained results show an increasing concentration of phenolic compounds, which means that comfrey root is rich in bioactive compounds. In this regard, these results are strongly correlated with the histological changes induced by the comfrey cream, which will be presented later.

#### 2.1.2. Cell Viability of Comfrey Extract

HaCaT is a commonly used aneuploid immortal keratinocyte cell line derived from adult human skin that underwent spontaneous transformation [35,36]. Considering that HaCaT cell viability is influenced by phenolic content, the concentration of the SyOf was calculated according to the total phenolic content expressed in μg/mL caffeic acid equivalent (382.69 µg/mL, 765.38 µg/mL, 1530.76 µg/mL, 3061.52 µg/mL, and 6123.05 µg/mL). No significant differences were observed between treated and untreated cells when treating the HaCaT cell line with different concentrations of SyOf ethanol extract (Figure 3). Therefore, SyOf ethanolic extract is not toxic to these human epidermal cells.

#### 2.1.3. Antioxidant Activity of Comfrey Extract

It is known that overproduction of free radicals exists due to endogenous and exogenous stress, which leads to various diseases. To evaluate the antioxidant activity of the extract, CUPRAC and FRAP assays were used (Figure 4). The formation of the colored copper chelate complex in the CUPRAC assay proved to be dependent on the antioxidant concentration in the SyOf extract, with a calculated sample EC50 of 269.5 µM Trolox/g (Figure 4A). The reduction potential of the ferric complex 2,4,6-tris(2-pyridyl)-1,3,5-triazine [Fe (III)-TPTZ] by the antioxidants found in Symphytum officinale L. root was dependent on the dose used in the experiment, with an IC50 of 324.1 µM Fe^2+^/g sample (Figure 4B). A FRAP value, in the same range interval, of 0.274 ± 0.003 mM/g was previously reported for Symphytum officinale root extract [37]. It is possible that the antioxidant activity of the extract came from the SAs and RA found in SyOf extract, which are known for their great free radical scavenging activity [38,39,40].

#### 2.1.4. Antibacterial Activity of Comfrey Extract

The antibacterial activity of SyOf extract was evaluated using *Staphylococcus aureus*, *Staphylococcus aureus MRSA*, *Escherichia coli*, and *Pseudomonas aeruginosa*. Using the agar diffusion test, no antimicrobial effect was observed, probably due to the low diffusibility potential of the extract in MH agar and the color of the extract, respectively. Consequently, the microdilution method was performed where the results were differentiated according to the concentration of the product (calculated based on total phenols) and according to the bacterial strain, respectively. MIC index results are shown in Table 2. The obtained results demonstrated that the ethanolic extract of SyOf has antibacterial action. Therefore, once this cream is used, no other antibiotics are used against possible infections that may appear in such situations.

### 2.2. Characterization and In Vivo Evaluation of Oil-in-Water Cream with Comfrey Extract

#### 2.2.1. Creams Characterization

Viscosity is an important property for topically applied products, as it influences their stability and their ability to sample from the packaging, to spread on the skin, to hold in place and even their sensory attributes. As can be seen from Figure 5A, the cream is a non-Newtonian fluid, with a thixotropic behavior, as the viscosity decreases when the shear rate increases. The extent of the thixotropy can be appreciated from the flow curve. Thus, the down curve obtained when the shear rates decrease follows the up curve but not very closely, and a small hysteresis area is formed. This behavior may be attributed to polyacrylamide—a gel former that is included in the cream formula and ensures easy spreading on the skin and a rapid recovery of the initial viscosity after application.

Figure 5B presents a rheogram of the SyOf cream, where the shear stress increases when the shear rate increases. Analysis of the plot according to the Bingham model allowed for the calculation of yield stress, which was 846.4 D/cm^2^. Thus, a low force is needed to remove the cream from a tube or a jar, which confirms the ease of sampling and spreading of the cream. The cream can hold a shape, but it is easily spread when shear stress is applied, a behavior suitable for application to injured skin.

The presence of SAs and RA in the creams with SyOf extract are confirmed by UV-Vis and FT-IR spectra (Figure 6). The specific absorption bands of SAs and RA are visible at 236 and 304 nm (Figure 6A). The FT-IR spectra of the creams revealed the presence of macadamia nut oil and glycyrrhiza glabra (Figure 6B). The absorption band at 1746 cm^−1^ can be assigned to the stretching vibration of the C=O group from macadamia nut oil [41] and to the b=vibration of the C=O group from glycyrrhizin, the main active compound of glycyrrhiza glabra [42]. The band at 1648 cm^−1^ can be assigned to the vibration of the C=C bond from glycyrrhizin [42]. The band at 1648 cm^−1^ originated from the stretching vibration of C=O groups from polyacrylamide [43], the main component of Sepigel 305^®^. The absorption band at 1038 cm^−1^ can be assigned to the stretching vibration of the C=O group from comfrey and the C-O stretching vibration from glycerin. The intensity ratio of the 1648 and 1038 cm^−1^ bands decreases with the addition of SyOf in the cream, from 1.72 to 1.62 in the case of SyOf 10% and from 1.72 to 1.1 in the case of SyOf 20%. This indicates an increase in intensity of the 1038 cm^−1^ absorption band, indicating the presence of comfrey in the prepared creams.

#### 2.2.2. Epithelium Bacteriological Examination

Pathogenic bacterial microflora can influence the process of skin drainage and regeneration. A bacteriological examination was carried out before skin defects were induced to identify whether there was a bacterial load on the skin that could influence the healing and regeneration process. After incubation, macroscopic examination showed that round, smooth, small- to medium-sized, cretaceous white-pigmented, non-hemolytic colonies developed on the seeding medium (Figure 7A,B). Smears were taken from the grown colonies and stained by the Gram method and examined microscopically. Microscopic examination identified a bacterial germ of the genus *Staphylococcus intermedius*, which is considered a saprophytic skin germ.

In conclusion, we can state that no pathogenic bacterial microflora was identified on the skin that could have influenced the process of vacuuming and regeneration; the bacterial flora that was identified was commensal, with a protective barrier role.

#### 2.2.3. Macroscopic Examination of Wound Healing

The distinctive feature of this study was the decision to use a macroscopic analysis of the region of interest (ROI), as opposed to a statistical analysis, since the latter would be useless given that the ROI changes over time. The macroscopic investigation of the skin’s healing reveals that the SyOf 20% cream-treated skin defects regenerate more quickly than the SyOf 10% cream-treated skin flaws, which take 12 days to regenerate (Figure 8 and Figure 9 and Table 3). The allantoin and RA from comfrey may be responsible for the faster wound healing when using SyOf 20% cream. It has been demonstrated that the RA from comfrey inhibits complement activation both in vivo and in vitro [6], but it is known that the biological activity of comfrey extract can be attributed to the interaction of different active compounds in the extract [44]. At the site of inflammation where complement activation is occurring, RA suppresses complement activation by covalently reacting with the activated complement component C3b without having the negative side effects of other medications like glucocorticoids and anti-inflammatory drugs. The pro-inflammatory gene cyclooxygenase 2 (COX2) may be strongly inhibited by RA [6].

#### 2.2.4. Histological Analyses

Histological sections were carried out after 8 and 14 days in terms of evaluating the SyOf regeneration efficiency.

In the histological sections from the cream control group treated with Simple C after 8 days, the cutaneous defect is filled with granulation tissue that is covered at its surface by crust and on the marginal zones of the wound by surface epithelium. The superficial part of the granulation tissue is highly vascularized and highly populated with fibroblasts and scattered inflammatory cells (neutrophils, macrophages, and multinucleated giant cells with foamy cytoplasm). A low amount of collagen fibers was detected here. The deeper part of the granulation tissue is less cellularized by fibroblasts and fibrocytes and isolated inflammatory cells can be detected, including neutrophils, macrophages/multinucleated giant cells, and lymphocytes. The fibers, basically collagen, are more prominent in this region as compared to the superficial zone of the granulation tissue. The superficial epidermis is acanthotic in the marginal zones of the granulation tissue and overlaps with the peripheral zones of the scar tissue, suggesting the re-epithelization of the skin (Figure 10A).

In the histological sections from the SyOf 10% group after 8 days, the cutaneous defect is fully filled with granulation tissue that practically regenerated all dermis and hypodermis. The granulation tissue is highly vascularized and intensely cellularized by numerous fibroblasts and scattered fibrocytes, whereas the collagen fibers are discrete. The inflammatory cells (basically, neutrophils and macrophages) can be detected on the superficial part of the granulation tissue, at the junction with the crust, which is prominent in the central part of the defect. Some hemorrhages along with inflammatory cells can be detected just below the crust. The skin re-epithelization is visible on the margins of the defect, whereas the central zone of the defect still has the cell debris of the crust (Figure 10B).

In the histological sections from the SyOf 20% group after 8 days, the skin regeneration is complete, including the epidermis and dermis/hypodermis. The newly made epidermis entirely covers the area of defect and displays acanthosis and cellular spongiosis. In the dermis, the granulation tissue is intensely cellularized by fibroblasts and a limited number of fibrocytes. The inflammatory infiltration is represented by scattered neutrophils and macrophages in the marginal zone, which are detected groups of multinucleated giant cells belonging to the macrophage line. In the mass of the granulation tissue, isolated clefts with a translucent material can be visualized, bordered by macrophages (Figure 10C).

Histological examination of the sections from the control group treated with Simple C cream, after 14 days, showed that the surgical wound was completely healed, and the epidermis was continuous. The superficial dermis and deep dermis contain many pilosebaceous units (Figure 11A,B,E,F). Occasionally, within the deeper dermis and hypodermis, a few macrophages admixed with rare lymphocytes and plasma cells are present (Figure 11D,H). The panniculus carnosus is present in these sections (black star, Figure 11A,B).

In the histological sections from the SyOf 10% group after 14 days, the surgical wound was completely epithelialized, and the pilosebaceous units partly regenerated, especially at the margins of the defect (thin arrow, Figure 12B,C,F,G). The superficial dermis and deep dermis were replaced by a large amount of dense fibrous connective tissue (scar tissue), rich in thick bundles of collagen fibers oriented parallel to the epidermis (black stars, Figure 12C,G). The deep dermis was infiltrated by a moderate number of lymphocytes, plasma cells, and macrophages (Figure 12D,H). An amorphous, basophilic foreign material is observed in the deep dermis (thick arrow, Figure 12D), without a major inflammatory reaction.

In the histological sections from the SyOf 20% group after 14 days, the surgical wound was healed, covered by a regular epidermis. The pilosebaceous units were present, and the panniculus carnosus was intact (Figure 13A–D). In the deep dermis, numerous macrophages, plasma cells, and a few eosinophils were present (black star, Figure 13D). An amorphous, basophilic foreign material (interpreted as the test material) was present in the cytoplasm of a few macrophages (Figure 13D).

The obtained histological analysis is summarized in Appendix A, for comparison purposes. In conclusion, after only 8 days, following the treatment with SyOf 10% cream, all the dermis and hypodermis were regenerated. The margins of the defect show evidence of skin re-epithelialization; however, the central area of the defect still contains crust cell debris. Furthermore, the epidermis, dermis, and hypodermis of the skin had fully regenerated after 14 days. The cutaneous defect is filled with granulation tissue in the histological sections from the cream control group, which is crusted at the surface and covered by surface epithelium in the wound’s marginal zones. Neutrophils, macrophages, and multinucleated giant cells with foamy cytoplasm are scattered across the superficial area of the granulation tissue, which is also densely populated with fibroblasts. Here, a small number of collagen fibers were found. These results are in agreement with those obtained by Dähnhardt et al. [45]. Within 4–7 days after using comfrey cream, skin cells began to regenerate more quickly and began to differentiate sooner toward a normal fine structure with discrete epidermal strata, keratin, and corneocyte production. This study used foreskin samples taken from 2- to 5-year-old donors’ foreskins compared to our study, which used rats. The entire extract, rather than just a few isolated ingredients that might be influencing the overall impact, is defined as the active component, as is typically the case with herbal therapeutic treatments. For instance, the phytochemical components of comfrey are known to include allantoin, RA, and SAs, which are recognized for their anti-inflammatory and regenerative effects. Moreover, in patients with acute coronary syndrome (ACS), the comfrey ointment sped up the healing of bruises brought on by enoxaparin injections [46]. In addition, the study by Lin and his coworkers suggests that SAs promote autophagy, which increases the survival of random-pattern skin flaps, and that this promotes angiogenesis, apoptosis, and oxidative stress [47]. Microemulsion containing SA B reduced acanthosis, lessened disease severity, and inhibited interleukin-23/interleukin-17 (IL-23/IL-17) cytokines, epidermal proliferation, and enhanced skin moisture in mice. Based on this, SAs might represent a potential new therapeutic medication for the treatment of psoriasis [48]. Other research has shown similar changes to ours in the treatment of RA. In this sense, the group of rats that received RA treatment had wounds that had a more advanced re-epithelialization and a better arrangement of the collagen bundle [9].

This study has some limitations, such as a small number of individuals and the fact that we did not examine toxicity in the most important organs. Even though the organ topology in rats is very similar to that of humans, we do not know the impact of extrapolating the results to humans.

## 3. Materials and Methods

### 3.1. Preparation and Characterization of Comfrey Extract

#### 3.1.1. Preparation of Extract from Comfrey Roots

The raw material (Symphytum radix of *Symphytum officinale* L.) was harvested in February 2022 in the Vâlcea area along the Olt River, Romania. The root was dried and powdered, and 50 g of it was used to obtain a polyphenolic extract by refluxing three times with 65% ethanol (*v*/*v*) for 30 min at 60 °C. The extract was evaporated to dryness by using a rotavapor. The comfrey concentrated extract (SyOf) was finally dissolved in water and then stored at −20 °C until analysis.

#### 3.1.2. Determination of Total Polyphenols Content of Comfrey Extract

The total phenolic content of SyOf was determined by the Folin–Ciocalteu spectrophotometric method with minor adaptations as described by Singleton [49]. Briefly, a 25 μL sample (extract) was mixed with 1800 μL of distilled water, 120 μL of Folin–Ciocalteu reagent, and 340 μL of Na_2_CO_3_ (7.5% in water). After 60 min of incubation in the dark at room temperature, the absorbance of the samples was measured in a Microplate Reader (BioTek Instruments, Bad Friedrichshall, Germany) at a wavelength of 750 nm. Results were expressed as mg of caffeic acid equivalents/100 g SyOf [50].

UV-Vis absorption spectra were recorded using a Jasco V-780 UV-Vis spectrometer (Jasco, Tokyo, Japan) with a 1 nm spectral resolution, using water as solvent. The FT-IR absorption spectra were recorded with a Jasco 6200 FT-IR spectrometer (Jasco, Tokyo, Japan) within the range of 400–4000 cm^−1^ and a spectra resolution of 4 cm^−1^ by using the KBr pellet technique.

#### 3.1.3. Determination of Phenolic Compounds of Comfrey Extract

The phenolic compounds of SyOf were determined by LC-ESI^+^-MS method. Agilent 1200 HPLC system equipped with a quaternary pump, solvent degasser, autosampler, UV-Vis photodiode detector (DAD) coupled with Agilent single quadrupole mass detector (MS) model 6110 (Agilent Technologies, Santa Clara, CA, USA) was used. Compound separation was performed on a Kinetex XB C18-100 Å column (4.6 × 150 mm, 5 μm particles (Phenomenex, Torrance, CA, USA) using the following mobile phases: A, water + 0.1% acetic acid; B, acetonitrile + 0.1% acetic acid. The gradient below was used for 30 min at 25 °C with a flow rate of 0.5 mL/min. Gradient (expressed in % B): 0 min, 5% B; 0–2 min, 5% B; 2–18 min, 5–40% B; 18–20 min, 40–90% B; 20–24 min, 90% B; 24–25 min, 90–5% B; 25–30 min, 5% B. All bend values fell between 200 and 600 nm. Chromatograms were recorded at wavelengths λ = 280 and 340 nm. For the MS, the ESI-positive ionization mode was used under the following working conditions: 3000 V of capillary voltage, 3500 °C, 7 L/min of nitrogen flow rate, 120–1200 *m*/*z*, full-scan, 35 psi of nebulizer pressure. Agilent ChemStation software 6110 was used to collect data.

#### 3.1.4. Determination of Allantoin Compounds of Comfrey Extract

Analysis was carried out using Agilent-1200 liquid chromatograph equipped with a degasser for solvents, quaternary pump, manual injector, and VWD detector (Agilent-Techonologies, CA, USA). The separation of allantoin was performed on a reverse-phase column Luna C18 250 × 4.6 mm, 5 µm particle size (Phenomenex, CA, USA). Acetonitrile and potassium dihydrogen phosphate 50 mM solution in a ratio of 20:80 at pH = 3.5 was used as the mobile phase. The flow rate was maintained at 0.8 mL/min, column temperature was T = 25 °C, and the detection was performed at 200 nm. Data acquisition was performed with the OpenLab–ChemStation (Agilent Techologies, CA, USA), Rev C.01.09 [144] version.

The comfrey extract was diluted with distilled water, filtered through a nylon filter (Chromafil Xtra PA-45/13 0.45 µm), and 20 μL was injected into the HPLC system. The allantoin content was determined using a five-point calibration curve of allantoin (R^2^ = 0.9983) in the linearity range 100–1000 µg/mL.

Acetonitrile (HPLC-gradient) and potassium dihydrogen phosphate were provided by Merck (Darmstadt, Germany), and water was purified with a Direct-Q UV system by Millipore (Bedford, MA, USA). Standard allantoin (purity ≥ 98%) was purchased from Merck (Germany).

#### 3.1.5. Cytotoxicity of the Comfrey Extract

The human keratinocytes (HaCaT) cell line was kindly provided by the Radiotherapy, Radiobiology, and Tumoral Biology Laboratory of the “Ion Chiricuţă” Institute of Oncology Cluj-Napoca, Romania. HaCaT cells were maintained in DMEM culture medium supplemented with 10% fetal bovine serum and 1% antibiotic–antimycotic at 37 °C in a humidified atmosphere with 5% CO_2_. To evaluate the cytotoxic potential of SyOf extract on HaCaT cells, a concentration of 1 × 10^5^ cells was seeded in 96-well tissue culture plates in a normal propagation medium. After 24 h of incubation, the HaCaT cells were treated with different concentrations of SyOf and incubated at 37 °C in a humidified atmosphere supplemented with 5% CO_2_ [51]. The concentration of the SyOf was calculated according to the total phenolic content expressed in μg/mL caffeic acid equivalent (382.69 µg/mL, 765.38 µg/mL, 1530.76 µg/mL, 3061.52 µg/mL, and 6123.05 µg/mL).

Untreated cells (cells kept in a typical propagation medium) served as the negative control. Cell viability was measured using CCK-8 assay following the manufacturer’s protocol. For this purpose, after 24 CCK-8 solution was added to each well and incubated for an additional 1.5 h, the optical density was subsequently determined at 450 nm by using a BioTek Synergy 2 microplate reader (Winooski, VT, USA). The obtained results were expressed as a viability percentage relative to the negative control (untreated cells) [51].

#### 3.1.6. Antioxidant Activity of Comfrey Extract

The cupric-ion-reducing antioxidant capacity (CUPRAC) of the extract was determined according to the protocol described previously by R. Apak et al. [52]. The absorbance was read at 450 nm against the blank with a JASCO V-630 spectrophotometer (International Co., Ltd., Tokyo, Japan). The antioxidant activity was expressed in µM Trolox equivalents, based on the calibration curve y = 0.0025x + 0.0358, R^2^ = 0.9448.

FRAP assay was used to monitor the extract’s ability to reduce ferric 2,4,6-tris(2-pyridyl)-1,3,5-triazine [Fe (III)-TPTZ] complex to the intensely blue-colored ferrous complex [Fe^2+^-(TPTZ)_2_]^2+^, in acidic medium [53]. FRAP reagent is a mixture of 10 mM/L TPTZ with 40 mM/L ferric chloride in acetate buffer (pH 3.6). FRAP reagent (180 μL) was mixed with extract (20 μL). The absorbance of the colored complex was read at 593 nm after 30 min of incubation at 37 °C. Results were calculated and expressed as μM Fe^2+^/g extract, based on the calibration curve y = 1.603x − 0.0014, R^2^ = 0.9999.

#### 3.1.7. Antimicrobial Assay on Comfrey Extract

The in vitro antimicrobial potential of ethanolic extract of SyOf (65% *v*/*v*) was tested by the agar diffusion test (Kirby–Bauer), according to the European Committee on Antimicrobial Susceptibility Testing (EUCAST) guidelines [54]. Four reference bacterial strains (n = 4), methicillin-susceptible *Staphylococcus aureus* ATCC 25923, methicillin-resistant *Staphylococcus aureus* ATCC 700699 (MRSA), *Escherichia coli* ATCC 25922, and *Pseudomonas aeruginosa* ATCC 27853, were used for this purpose. Overnight bacterial suspensions at a concentration of McFarland 0.5 were used for testing. Mueller–Hinton (MH) agar plates were inoculated (overnight culture) by flooding. After drying the plate surface, wells of 6 mm diameter were cut equidistantly, after which tested solutions were added. The concentration of the SyOf was calculated according to the total phenolic content expressed in μg/mL caffeic acid equivalent. Gentamicin discs (10 µg) were used for the reference control. Testing was performed in triplicate. Results were evaluated after 24 h of incubation at 37 °C by measuring the diameters of the growth inhibition zones.

The minimum inhibitory concentrations (MICs) were assessed using the microdilution method according to Clinical and Laboratory Standards Institute (CLSI) guidelines (2018) [55]. The evaluation was performed using the broth microdilution (2-fold dilution) method on 96-well plates, in triplicate. Briefly, 100 µL of MH broth was added to each well of the 96-well plates, and SyOf ethanolic extract and 20 µL of bacterial suspension (1.5 × 10^6^ CFU/mL) were also added in each well. The plates were incubated at 37 °C for 18 h. For bacterial growth/inhibition, after 18 h of incubation, 20 µL of MTT solution 3-(4,5-dimethylthiazol-2-yl)-2,5-diphenyltetrazolium bromide, 1.25 mg/mL) was added to each well. Cultures were incubated for 1 h at 37 °C, and bacterial growth was indicated by the appearance of a chromogenic reaction (purple color) and inhibition of growth by clear/yellow staining in the wells. Testing was performed in triplicate. The lowest concentration that visibly inhibited bacterial growth was defined as MIC. The results were interpreted in comparison with the untreated control culture (MH broth) [56].

The minimum bactericidal concentration (MBC) value, which represents the lowest concentration that prevents bacterial growth, was also assessed. To evaluate MBC values, 100 µL of bacterial suspension was collected from the well where no visible bacterial growth was observed. The suspensions were inoculated on MH agar plates and incubated for 18 h at 37 °C. The MIC index was also calculated, based on the MBC/MIC ratio. Thus, an MBC/MIC ratio ≤ 4 was considered bacteriostatic, while an MBC/MIC ratio ≥ 4 was regarded as a bactericide.

### 3.2. Preparation and Characterization of Oil-in-Water Cream of Comfrey Extract

#### 3.2.1. Preparation of the Oil-in-Water Cream with Comfrey Extract

The cream was prepared as an oil-in-water (O/W) disperse system using the following ingredients: caprylic/capric triglycerides (4%, *w*/*w*; Croda, Snaith, UK), macadamia integrifolia nut oil (5%, *w*/*w*), *Glycyrrhiza glabra* extract, glycerin (5%, *w*/*w*; Elemental, Oradea, Romania), paraffin oil (6%, *w*/*w*), cetearyl alcohol (6%, *w*/*w*; Vitamar, Bucarest, Romania), Sepigel 305^®^ (3.5%, *w*/*w*; polyacrylamide and C13-14 Isoparaffin and laureth-7, Seppic, Paris, France), Euxyl PE 9010^®^ (0.5%, *w*/*w*; phenoxyethanol and ethylhexylglycerin, Schülke & Mayr, Norderstedt, Germany), and distilled water (to 100%, *w*/*w*). SyOf was selected as the active ingredient of the O/W cream, at concentrations of 10% and 20% (*w*/*w*; SyOf 10% and SyOf 20%), as it controls the inflammatory process and induces collagen deposition (1). Also, macadamia nut oil acts as an emollient and regenerative through its content of monounsaturated acids and contains antioxidants that can reduce inflammation and oxidative stress in the skin (2), while *Glycyrrhiza glabra* has anti-inflammatory activity (3) and was added at a concentration of 1%.

First, the aqueous phase was made by mixing distilled water at a controlled temperature (50 ± 2 °C) together with Sepigel 305 ^®^, glycerol, and the preservative Euxyl PE 9010^®^. Sepigel 305 creates a gel texture with distilled water, being a multifunctional vehicle with thickening, stabilizing, texturizing, and tissue-adhering properties (4). Separately, the paraffin oil was placed in a water bath at 60 ± 2 °C, together with cetearyl alcohol. This lipophilic melted mixture was added to the water phase at 50 ± 2 °C under continuous stirring. After removing this mixture from the water bath, when the previous mixture reached a temperature below 40 °C, macadamia nut oil and caprylic/capric triglycerides were added under continuous stirring, which was continued until the cream reached room temperature. Then, the *Glycyrrhiza glabra* extract and the SyOf were both homogeneously incorporated into the cream composition. For the control, the cream without SyOf (Simple C) was prepared.

#### 3.2.2. Characterization of the of the Oil-in-Water Cream with Comfrey Extract

The cream viscosity was determined using a cone-and-plate viscometer (CAP 2000+, Brookfield, Phoenix, AZ, USA). The cream was applied to the flat surface of the viscometer, the disc was sealed, and the results were recorded, using the following viscometer settings: hold time 10 s, run time 30 s, spindle 08 at rotations speeds varying from 5 to 50 RPM. Measurements were performed in triplicate, at 23 ± 1 °C, and the mean value ± standard deviation was reported.

The obtained creams were characterized by UV-Vis and FT-IR spectroscopy using the equipment described above.

### 3.3. In Vivo Evaluation of the Oil-in-Water Cream with Comfrey Extract

#### 3.3.1. Animal Care and Use

In this research, the biological material used was represented by 15 rats, family *Muridae*, Wistar-Lewis line, female, weighing 150–180 g. Docility to experimental maneuvers, reduced susceptibility to bacterial infection, low degree of spontaneous tumor development, and good adaptability to captive rearing were the arguments underlying the choice of the animal model [57]. Rats were purchased from the Experimental Medicine Centre of the Iuliu Hațieganu University of Medicine and Pharmacy, Cluj-Napoca, Romania, and housed and maintained at the Unit for Reproduction and Use of Laboratory Animals of the Faculty of Veterinary Medicine, Cluj-Napoca, where the experiment was conducted. Maintenance and feeding conditions were standardized for all rats in the study, ensuring a temperature of 23 °C, humidity cycles of 55%, and light/dark cycles of 12 h, according to [58], and the feed administered consisted of standard granulated rodent food. The rats’ access to food and water was ad libitum.Three batches were formed to experiment on, each batch having 5 rats. The experiment was approved by the Bioethics Committee of the University of Agricultural Sciences and Veterinary Medicine, Cluj-Napoca, no. 245/05.04/2021 and authorized by the Sanitary-Veterinary and Food Safety Directorate, Cluj-Napoca, by Project Authorization no. 256/13 May 2021.

According to the legislation in force, obtaining bioethics approvals and Sanitary-Veterinary and Food Safety Directorate authorization is a mandatory practice in conducting in vivo studies on animals. The results indicating good regeneration in the case of the cream with SyOf justifies running the experiment.

#### 3.3.2. Surgical Procedure

The actual surgery was performed only after each rat was weighed and anesthetized with a mixture of Ketamine and Xylazine (Xylazin Bio 2%, Bioveta, Czech Republic, 6 mg/kg and Ketamine Narkamon Bio, Bioveta, Czech Republic 60 mg/kg), injected intraperitoneally [59]. After anesthesia, the excision site was prepared for surgery by trimming the hair in the dorsal thoracic area from T1 to L1, and special attention was paid to the skin from the region of the withers between the two scapular joints to the level of the pelvis. After trimming, a depilatory cream was applied to ensure complete hair removal and left to work for 3 min to avoid burning, and then the depilatory cream was removed with a special plastic scraper. Antisepsis of the area was achieved by cleaning it with sterile swabs soaked in a Lifo-Scrub solution (Braun Medical), after which sterile swabs soaked in 70% sanitary alcohol were used to achieve a better antiseptic effect. Four dermal excisions were performed in the body area prepared for the intervention, following the following steps: the rat was fixed in a lateral position, and then the skin in the dorsal region, along the line of the spine, was clamped and pulled with two hemostatic tweezers, obtaining a skin fold. Dermal excisions were performed using a biopsy punch, fixed at a distance of approximately 8 mm from the edge of the skin fold. The 4 dermal excisions were obtained by gently rotating the biopsy punch through both layers of the skinfold, the distance between the 2 excisions being approximately 16 mm as a result of the skinfold unraveling [60]. After dermal excisions were performed, they were cleaned with a sterile swab. To minimize the constricting effect of the panniculus carnosus muscle on each excision, a silica ring was applied. Superglue was applied to the side of the silica ring that came into contact with the skin for immediate fixation, which then allowed them to be sutured with non-absorbable sutures so that each silica ring was fixed to the skin with 4 symmetrical sutures [60]. For the application of the cream with the comfrey extracts, the rats were restrained in sterno-abdominal decubitus, with the hind limbs oriented to the right of the researcher, so that the dorsal region with the 4 excisions remained free for the application of the experimental product. The dermal excision on the lower left was the control excision for all 3 batches. A simple cream without any extract was applied to it. The dermal excision on the upper left side was the excision on which the experimental product was applied, i.e., SyOf 10% cream and SyOf 20% comfrey extract cream were applied to the dermal excision on the right side. Nothing was applied to the dermal excision on the lower right side, considered the control excision, and regeneration occurred spontaneously. After the application of the simple cream (Simple C) and the experimental product, the rats were dressed in sterile elastic bandages to avoid wound infection. On the 4th and 7th days of the experiment, a new application of ointment was made, following the same protocol and in the same concentrations, after which the rats were bandaged, following the same steps. For 15 days, the rats in this study were monitored for health and skin healing and regeneration. On day 15 of the experiment, after complete wound healing, each rat was euthanized by anesthetic overdose and cervical dislocation. Samples of skin tissue were taken from the site of each dermal excision, covering both the area where the scar had formed and normal surrounding tissue of approximately 1 cm, for histopathological examination.

Materials used in the survey were: biopsy punch with a diameter of 5 mm—by relaxing the skin, the excision had a diameter of 6 mm; solutions for antisepsis such as Lifo-Scrub (Braun Medical) and sanitary alcohol 70%; scalpel with fixed blade; surgical tweezers and hemostatic tweezers; scissors; needles and non-absorbable sutures (Nyllion 4.0); clipper; depilatory cream (Veet, France); sterile dressing; elastic bandage and superglue.

#### 3.3.3. Epithelium Bacteriological Examination

The method used is a qualitative one. Samples were taken before skin defects from 3 rats. All rats included in this study received the same feeding, watering, and microclimate conditions. For bacteriological examination, several steps were followed, such as collection of biological samples, transport, seeding on special culture media, and identification of germs based on certain characteristics by performing macroscopic and microscopic examination. The sampling was carried out on the first day of surgery. The samples were taken from the skin before trimming, shaving, and disinfecting the area to see if there were pathogenic bacteria on the skin that could then influence the healing of the skin. After collection, samples were seeded in Petri dishes on blood-enriched agar and incubated at 37 °C for 24 h. The results indicate the absence of pathological bacteria and only the presence of normal ones. Also, in the absence of clinical symptoms such as scratching or hair loss, the animals were considered clinically healthy without pathological dermal factors influencing the healing of the skin during the experiment.

#### 3.3.4. Macroscopic Examination of Wound Size Reduction

The pictures of the wounds were taken on days 1, 4, 7, 12, and 14 and were used to quantify the degree of healing. For the evaluation of the images, free, open-source software was used (ImageJ 1.54g), and each picture was transformed into an 8-bit picture, grayscale, at dimensions of 336 × 336 pixels, to have homogeneity. Before image capture, wound dimensions were measured using a standard ruler, establishing a calibration of 27 pixels per millimeter within the software. A free-hand ROI was traced to include the wound and the changes in the ROI were evaluated. This ROI enabled the calculation of pixel value alterations throughout the healing timeline. Based on these selections, surface plots were generated to provide a visual representation of the healing process, comparing pixel intensity levels at the wound site against those of the surrounding intact skin surface. Thus, to have a graphical representation of the wound, a histogram was obtained for each ROI traced. This methodology was consistently applied across all specimens, ensuring a standardized approach for image capture and analysis. The same process was used for all products, and pictures were obtained.

#### 3.3.5. Histological Method

In the middle of the healing period (eight days), after fixation in 10% buffered formalin, the samples underwent dehydration with alcohol in increasing concentrations (70 O, 96 O, and 100 O), clarified with 1-butanol (three baths, 1 h each) and embedded in paraffin blocks. Eventually, 5 μm thick tissue sections were prepared using a LEICA RM2125RT microtome. Finally, tissue sections were stained by Goldner’s trichrome method for general histological aspects. The achieved histological slides were examined under an Olympus BX41 microscope, and the microphotographs were obtained by using a built-in Olympus U-TV0.35XC-2 T8 camera.

At the end of the experimental period, the animals were killed by cervical dislocation, and skin samples were harvested for histological examination. Skin samples were fixed in 10% buffered neutral formalin and embedded in paraffin, sections were made at 4 μm, and the slides were stained by Haematoxiline–Eosine (HE) and Masson’s trichrome methods. The histological sections were examined under an Olympus BX 51 microscope, images were taken using the Olympus UC 30 digital camera, and they were processed using an image acquisition and processing program: Olympus Stream Basic.

#### 3.3.6. Pain Management

The procedures involving the handling of animals were led following the guidelines of Directive 2010/63/EU, Romanian National Low 43/2014, and ISO 10993-2 Animal Welfare Requirements [58]. Rats were anesthetized in an Isoflurane chamber (Isoflutek 1000 mg/g, Laboratorios Karizoo S.A., Voorschoten, The Netherlands, 3%) and were maintained under inhalation anesthesia via facial mask for preparation before and during surgery. Preoperatively, the following were administered subcutaneously: 5 mL of NaCl 0.9% (NaCl 0.9% B. Braun, Melsungen, Germany) to prevent dehydration. Sterile acupuncture needles (0.16 × 20 mm (Acimut, Madrid, Spain)) with copper handles were inserted 5 mm into the depression on the dorsal midline between the seventh cervical vertebrae and first thoracic dorsal spinous processes, at acupoint GV-14 (Governing vessel) and between the seventh lumbar vertebrae and first sacral vertebrae, on the dorsal midline, at acupoint Bai-hui (“Hundred meetings point”, Figure 14). The stimulation between the two acupoints described before was performed with a constant current pulse for 50 min (15 min for induction, 30 min during the surgery, and 5 min after the surgery). We used an electroacupuncture machine made by Dr. Huisheng, a Xie-JM-3A, the stimuli were set at 40 Hz, and the current intensity was increased in a stepwise fashion until a muscle twitch was observed (~1–1.5 V) [60].

After the surgery, every third day we performed electroacupuncture for pain management and also to improve the healing process. Electroacupuncture reduces local inflammation and provides blood supply for the skin. Rats were observed daily until completion of the experiment, and the Grimace scale was used based on changes in a number of facial action units [60].

### 3.4. Statistical Analysis

All data reported in cell viability assay and wound regeneration are reported as the mean ± SD. The triplicate (n = 3) values obtained for cell viability were analyzed by two-way analysis of variance (ANOVA). Statistical significance was at *p* < 0.05 in all cases.

## 4. Conclusions

Comfrey root extract showed a rich content of polyphenolic compounds, especially SAs and RA, as well as a rich content of allantoin. Referring to the compounds identified in the analyzed extract and the literature, we can conclude that a large part of the positive effects is due to both polyphenolic and allantoin compounds. Good antioxidant activity was obtained for the extracts, which are known for their great free radical scavenging activity. The antimicrobial effect of comfrey extract on epithelial cell lines demonstrated health effects, having a bactericidal property on four reference bacterial strains: methicillin-sensitive *Staphylococcus aureus*, methicillin-resistant *Staphylococcus aureus*, *Escherichia coli*, and *Pseudomonas aeruginosa*. Histological analysis of the skin defects 14 days after the intervention has shown that in the skin excisions treated with SyOf 20% the healing is complete, and the pilosebaceous units and the panniculus carnosus muscle are completely formed compared to those treated with SyOf 10% cream concentration where in the dermis and hypodermis there are areas of inflammation, compared to the control excisions where in the deep segment there is an inflammatory reaction with macrophages and lymphocytes.

Thus, the obtained results confirm the wound regeneration effect of comfrey extract. Still, we cannot exclude the possible synergistic effect since macadamia nut oil and glycyrrhiza glabra also have wound regeneration abilities.

## Figures and Tables

**Figure 1 ijms-25-03099-f001:**
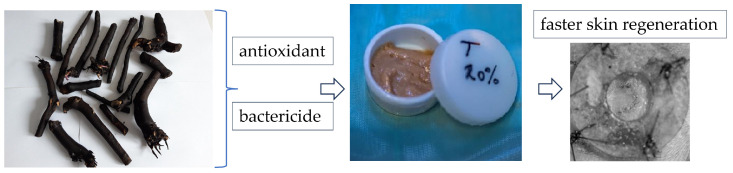
Schematic representation of comfrey extract’s investigation.

**Figure 2 ijms-25-03099-f002:**
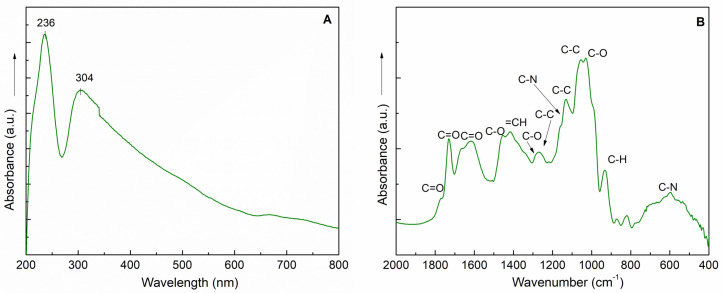
UV−Vis (**A**) and FT−IR (**B**) spectra of the SyOf extract.

**Figure 3 ijms-25-03099-f003:**
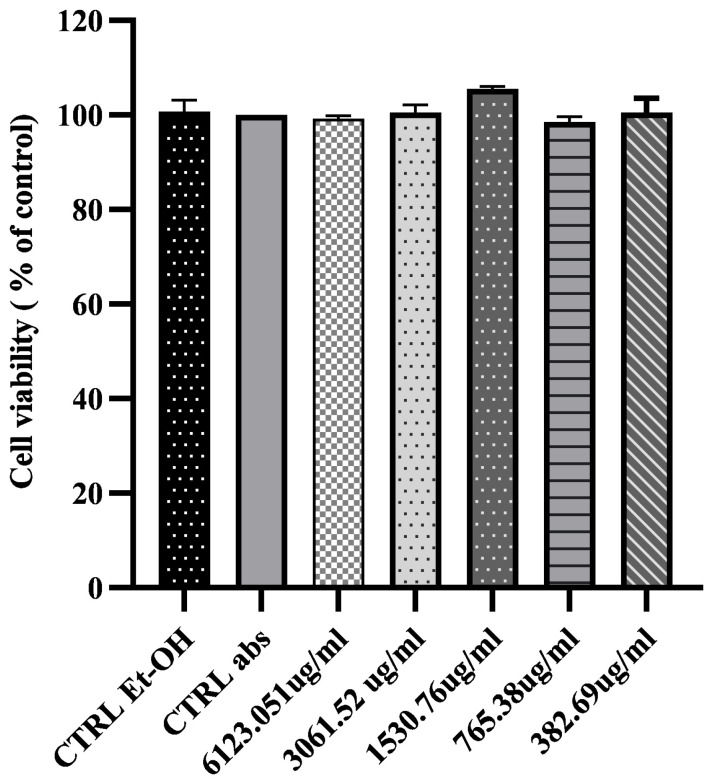
Proliferation kinetics of HaCaT cells following 24 h of treatment with different concentrations of SyOf ethanolic extract. The average cell vitality obtained after testing in triplicate was used to compare the results to the control (untreated cell cultures).

**Figure 4 ijms-25-03099-f004:**
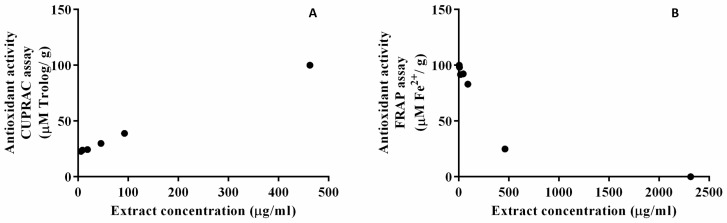
Antioxidant activity of the SyOf extract determined by CURAC (**A**) and FRAP (**B**) assays.

**Figure 5 ijms-25-03099-f005:**
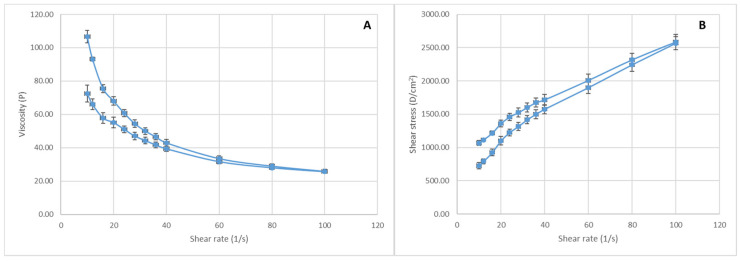
Flow curve (**A**) and rheogram (**B**) of the SyOf cream (mean values ± SD).

**Figure 6 ijms-25-03099-f006:**
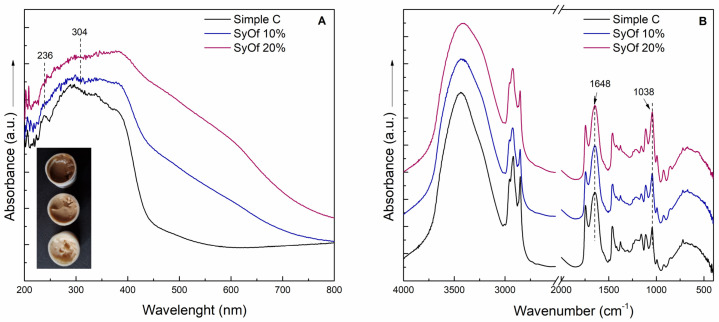
UV−Vis (**A**) and FT−IR (**B**) spectra of the Simple C, SyOf 10%, and SyOf 20% creams.

**Figure 7 ijms-25-03099-f007:**
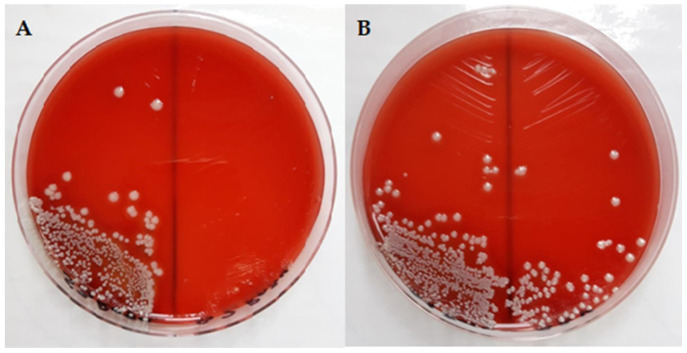
Representative macroscopic images of bacterial colonies grown on blood agar (**A**,**B**).

**Figure 8 ijms-25-03099-f008:**
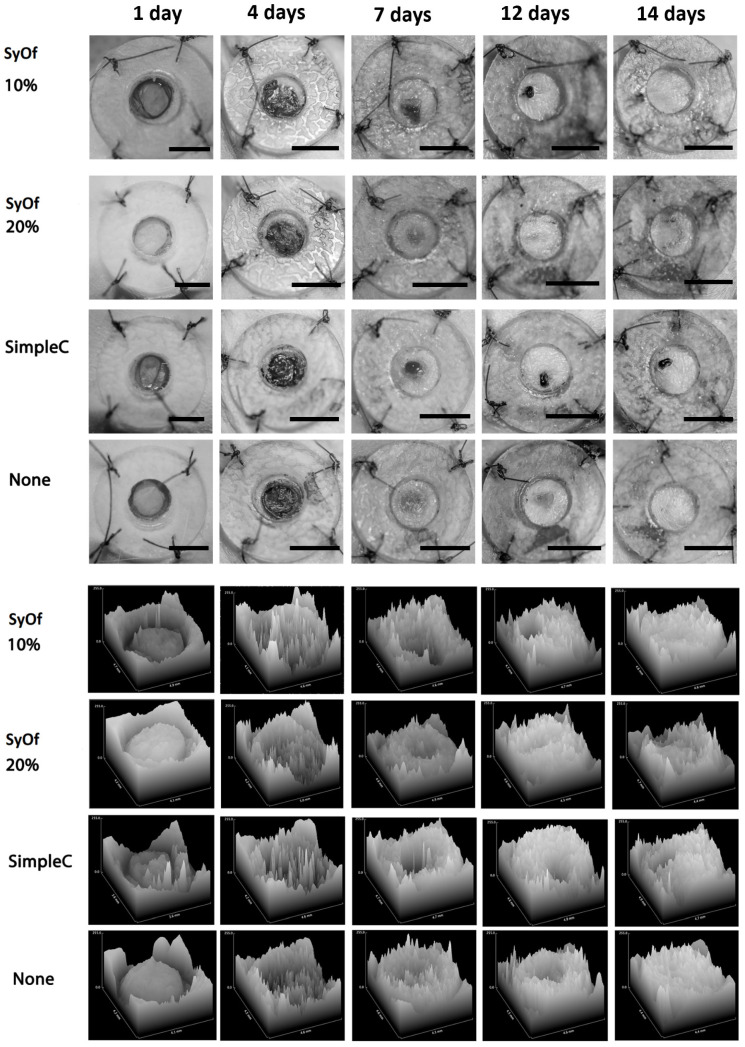
Full-thickness skin defect in rats and the evolution of wound healing treated with Simple C, SyOf 10%, and SyOf 20% cream and clear wound (none) presented in 3D at 1, 4, 7, 12, and 14 post-surgery days. Scale bars: 6 mm.

**Figure 9 ijms-25-03099-f009:**
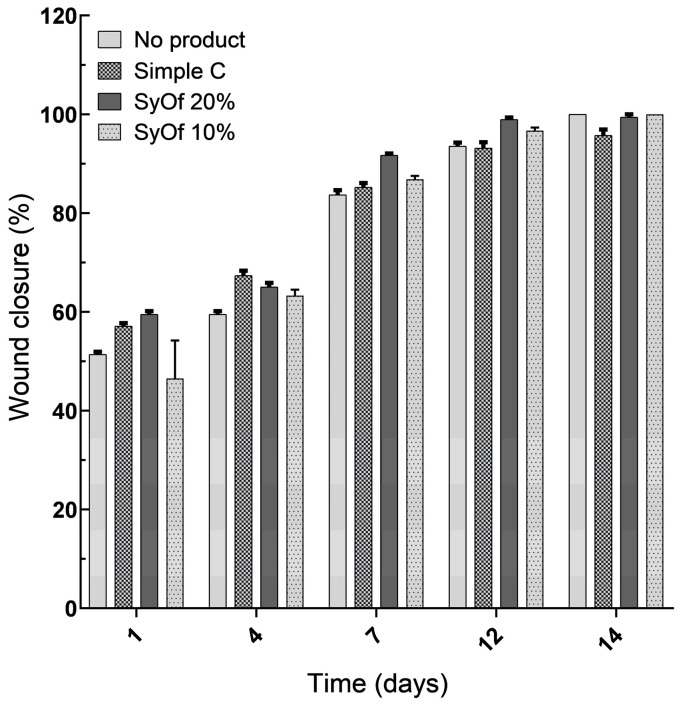
Percentage of wound closure for the defect treated with Simple C, SyOf 10%, and SyOf 20% cream and clear wound (no product). *p* < 0.05.

**Figure 10 ijms-25-03099-f010:**
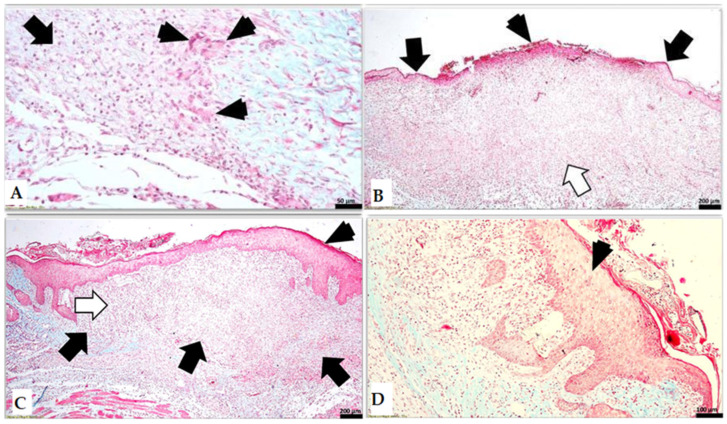
(**A**) (group with Simple C): the presence of inflammatory cells, including neutrophils, macrophages, and multinucleated giant cells (arrowheads) in the granulation tissue (arrow); (**B**) (group with SyOf 10%): the re-epithelization of the skin is visible on the margins of the formerly created defect (black arrows), whereas the central zone of the defect still has the crust (arrowhead) that is covering the granulation tissue (white arrow); (**C**) (group with SyOf 20%): the newly created epidermis (arrowhead) is entirely covering the area of the former defect (suggested by black arrows), whereas in the dermis the granulation tissue (white arrow) is intensely cellularized by fibroblasts, fibrocytes, and a limited number of inflammatory cells; (**D**) (group with SyOf 20%): acanthosis and cellular spongiosis (arrowhead). Scale bars: (**A**) 50 μm, (**B**,**C**) 200 μm, (**D**) 100 μm.

**Figure 11 ijms-25-03099-f011:**
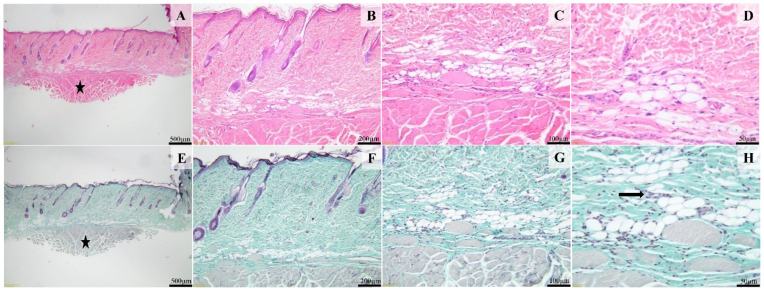
Histological images of the cream control group. The skin defect is completely healed, covered by an intact epidermis and dermis containing many pilosebaceous units. Focally, the fibrous reaction, especially within the deep segment, contains poorly demarcated foci consisting of a few macrophages admixed with rare lymphocytes and plasma cells (image (**H**), arrow). H&E stain (images (**A**–**D**)) and Masson’s trichrome (images (**E**–**H**)). Scale bares: (**A**,**E**) 500 μm, (**B**,**F**) 200 μm, (**C**,**G**) 100 μm, (**D**,**H**) 50 μm.

**Figure 12 ijms-25-03099-f012:**
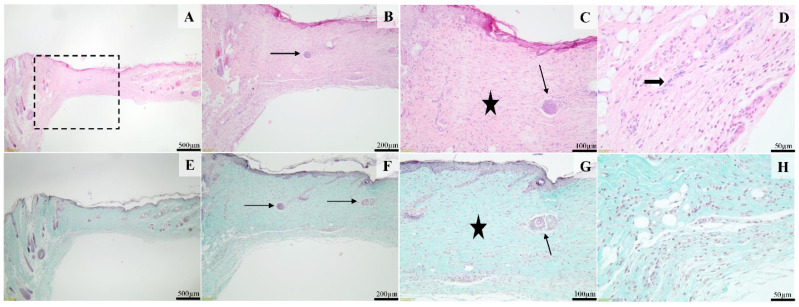
Histological images of the skin and subcutaneous tissue from the animals of the SyOf 10% group. The moderately hyperplastic epidermis is supported by dermal fibrosis (black star) containing few partly regenerated pilosebaceous units (thin arrows, (**B**,**C**,**F**,**G**)); Focally, intracellular amorphous, basophilic test material in the deep dermis (thick arrow, (**D**)) is present. H&E stain (images (**A**–**D**)) and Masson’s Trichrome (images (**E**–**H**)). Scale bares: (**A**,**E**) 500 μm, (**B**,**F**) 200 μm, (**C**,**G**) 100 μm, (**D**,**H**) 50 μm.

**Figure 13 ijms-25-03099-f013:**
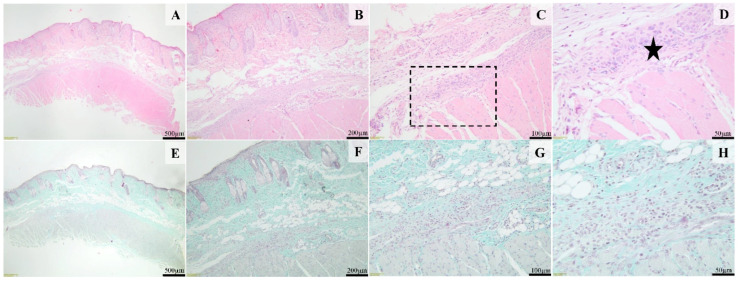
Histological images of the skin and subcutaneous tissue from the animals of the SyOf 20% group. The skin defect is covered by an intact epidermis. Multiple pilosebaceous units are present within the defect (**A**,**B**,**E**,**F**); a mild granulomatous reaction in the deep dermis centered on the test material (black star, (**D**)) was noted. H&E stain (images (**A**–**D**)) and Masson’s trichrome (images (**E**–**H**)). Scale bares: (**A**,**E**) 500 μm, (**B**,**F**) 200 μm, (**C**,**G**) 100 μm, (**D**,**H**) 50 μm.

**Figure 14 ijms-25-03099-f014:**
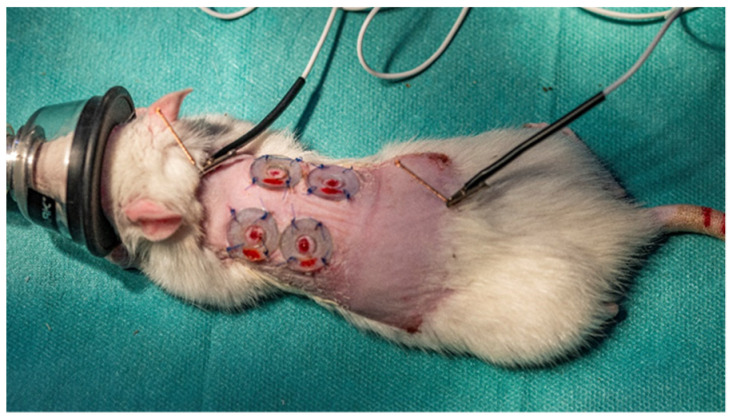
Moxibustion and acupuncture encourage the growth of fibroblasts and neoangiogenesis during the healing of experimental excisional wounds in adult female Wistar rats.

**Table 1 ijms-25-03099-t001:** Identification and concentration of phenolic compounds in the sample of comfrey extract, expressed in μg/mL caffeic acid equivalent.

Peak No.	R_t_ (min)	UV λ_max_ (nm)	[M + H]^+^ (*m*/*z*)	Compound	Subclass	μg/mL (Mean ± SD)
1	13.30	322	181	Caffeic acid	Hydroxycinnamic	123.162 ± 12.40
2	15.26	360, 240	539	Salvianolic acid I	Hydroxycinnamic	2382.23 ± 115.23
3	16.98	360, 250	719	Salvianolic acid B	Hydroxycinnamic	641.83 ± 51.34
4	17.87	330	361	Rosmarinic acid	Hydroxycinnamic	1055.02 ± 42.20
5	18.65	320, 260	495	Salvianolic acid A	Hydroxycinnamic	279.50 ± 27,93
6	19.60	320, 260	493	Salvianolic acid C	Hydroxycinnamic	1641.29 ± 80.06
				** *Total Phenolics* **		** *6123.051* **

**Table 2 ijms-25-03099-t002:** Assay for the bactericidal effect of the extract.

MIC Index MBC (μg/mL)/MIC (μg/mL)				
SyOf	*Staphylococcus aureus*	*Staphylococcus aureus MRSA*	*Escherichia coli*	*Pseudomonas aeruginosa*
	4 1530.76/382.69	2 1530.76/765.38	1 6123.01/6123.01	1 6123.01/6123.01

MIC, minimum inhibitory concentration; MBC, minimum bactericidal concentration; an MBC/MIC ratio of less than 4 was thought to be bacteriostatic, and an MBC/MIC ratio of greater than 4 was thought to be bactericidal.

**Table 3 ijms-25-03099-t003:** Comparative presentation of the macroscopic examination regarding SyOf 10%, SyOf 20%, Simple C, and clean wound (None).

Treatment	SyOf 10%	SyOf 20%	Simple C	None
	Healing of the wound starts to be visible on the 12th day and complete healing on the 14th day.	Healing of the wound starts to be visible on the 7th day and complete healing on the 12th day.	Healing of the wound starts to be visible from the 12th day, and healing is not complete on the 14th day.	Healing begins to be visible on day 7 and complete healing on day 14.

## Data Availability

Data are contained within the article.

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
