# Peer review of "Healing of Skin Wounds in Rats Using Creams Based on Symphytum Officinale Extract"

_ijms, 2024, doi:10.3390/ijms25063099_

Round 1
Reviewer 1 Report
Comments and Suggestions for Authors
The manuscript presents a study on the efficacy of creams containing Symphytum officinale (comfrey) extract in healing skin wounds in Wistar rats. The authors have conducted a series of experiments to evaluate the bacteriological, macroscopic, and histological aspects of wound healing. The study is well-structured and addresses an interesting topic within the field of wound healing and natural product research. However, there are several areas where the manuscript could be improved for clarity, scientific rigor, and completeness.
The description of the bacteriological examination (page 17) could be more detailed. It is unclear how the absence of pathogenic microflora was confirmed, and whether any quantitative methods were used to assess bacterial load.
The macroscopic examination of wound size reduction (page 17) would benefit from a more detailed explanation of the image analysis process. The authors should clarify how the free-hand ROI was standardized across different wounds and how the histogram analysis correlates with the healing process.
The histological method section (page 17) is well-described, but the results of the histological examination are not presented in a clear and concise manner. The authors should consider summarizing the key findings in a table or graph for easier interpretation.
The macroscopic examination of wound healing (page 8) mentions the use of SyOf 20% cream and its comparison with SyOf 10% cream. However, the results are not presented in a comparative format that would allow for a clear understanding of the differences in healing rates.
The manuscript lacks a clear presentation of statistical analysis. For a study of this nature, it is crucial to include statistical tests to determine the significance of the findings. The authors should provide p-values, confidence intervals, and other relevant statistical data to support their conclusions.
The discussion (page 12) provides a comparison with other studies and potential mechanisms of action for the comfrey extract. However, the authors should discuss the limitations of their study, such as the small sample size and the use of an animal model, which may not fully represent human wound healing.
The authors should also consider discussing the potential for translation of their findings to clinical practice, including any challenges that may arise in the process.
The use of abbreviations and technical terms should be consistent throughout the manuscript, and all abbreviations should be defined upon first use.
The authors mention ethical approval and authorization for the study (page 16). However, they should also discuss the ethical implications of their findings and any considerations for the use of comfrey extract in a clinical setting.
Comments on the Quality of English LanguageThe English quality of the manuscript requires minor editing
Author Response
REV1
The manuscript presents a study on the efficacy of creams containing Symphytum officinale (comfrey) extract in healing skin wounds in Wistar rats. The authors have conducted a series of experiments to evaluate the bacteriological, macroscopic, and histological aspects of wound healing. The study is well-structured and addresses an interesting topic within the field of wound healing and natural product research. However, there are several areas where the manuscript could be improved for clarity, scientific rigor, and completeness.
The description of the bacteriological examination (page 17) could be more detailed. It is unclear how the absence of pathogenic microflora was confirmed, and whether any quantitative methods were used to assess bacterial load.
A: The method used is a qualitative one, as we described in the working method, the samples were taken from the skin before trimming, shaving, and disinfecting the area to see if there are pathogenic bacteria on the skin that can then influence the healing of the skin. The results indicate the absence of pathological bacteria and only the presence of normal ones. Also, in the absence of clinical symptoms such as scratching or hair loss, the animals were considered clinically healthy without pathological dermal factors influencing the healing of the skin during the experiment.
The macroscopic examination of wound size reduction (page 17) would benefit from a more detailed explanation of the image analysis process. The authors should clarify how the free-hand ROI was standardized across different wounds and how the histogram analysis correlates with the healing process.
A: Photographs documenting the progression of wound healing were captured on days 1, 4, 7, 12, and 14 post-injury to facilitate a quantitative analysis of the healing process. Image analysis was conducted utilizing the open-source software ImageJ. To ensure uniformity across the dataset, each image was converted to an 8-bit grayscale format with dimensions of 336x336 pixels. Before image capture, wound dimensions were measured using a standard ruler, establishing a calibration of 27 pixels per millimeter within the software. To analyze changes in wound morphology, a free-hand oval Region of Interest (ROI) was delineated to encompass the wound margins accurately. This ROI enabled the calculation of pixel value alterations throughout the healing timeline. Based on these selections, surface plots were generated to provide a visual representation of the healing process, comparing pixel intensity levels at the wound site against those of the surrounding intact skin surface. This methodology was consistently applied across all specimens, ensuring a standardized approach for image capture and analysis.
The histological method section (page 17) is well-described, but the results of the histological examination are not presented in a clear and concise manner. The authors should consider summarizing the key findings in a table or graph for easier interpretation.
A: The obtained histological analysis is summarized in Table S1 (see supplementary materials), for comparison purposes.
The macroscopic examination of wound healing (page 8) mentions the use of SyOf 20% cream and its comparison with SyOf 10% cream. However, the results are not presented in a comparative format that would allow for a clear understanding of the differences in healing rates.
A: The comparative presentation of the macroscopic examination regarding the used creams was summarized in Table 3.
The manuscript lacks a clear presentation of statistical analysis. For a study of this nature, it is crucial to include statistical tests to determine the significance of the findings. The authors should provide p-values, confidence intervals, and other relevant statistical data to support their conclusions.
A: The manuscript was completed with statistical analysis.
The discussion (page 12) provides a comparison with other studies and potential mechanisms of action for the comfrey extract. However, the authors should discuss the limitations of their study, such as the small sample size and the use of an animal model, which may not fully represent human wound healing.
A: This study has some limitations, such as a small number of individuals and the fact that we did not examine toxicity in the most important organs. Even though the organ topology in rats is very similar to that of humans, we do not know the impact of extrapolating the results to humans.
The authors should also consider discussing the potential for translation of their findings to clinical practice, including any challenges that may arise in the process.
A: For acceptance in clinical practice (which does not necessarily have to be on humans but used in veterinary medicine), the study must be redone on a larger sample, on larger wounds for which it is necessary to apply the cream over a long-term period so that the in vivo toxicity on the organs can also be determined.
The use of abbreviations and technical terms should be consistent throughout the manuscript, and all abbreviations should be defined upon first use.
A: Thank you for your suggestions. The manuscript has been completed with abbreviations.
The authors mention ethical approval and authorization for the study (page 16). However, they should also discuss the ethical implications of their findings and any considerations for the use of comfrey extract in a clinical setting.
A: According to the legislation in force, obtaining bioethics approvals and Sanitary-Veterinary and Food Safety Directorate authorization is a mandatory practice in conducting in vivo studies on animals. The result indicates a good regeneration in the case of the cream with SyOf justifies the experiment's running.
Reviewer 2 Report
Comments and Suggestions for Authors
In the paper entitled "Healing of skin wounds in rats using creams based on Symphytum officinale extract", the Authors described the fabrication, characterization, and biological evaluation of polyphenolic comfrey root extract. The presented research is very insightful and worth publishing in IJMS. However, some minor issues have to be addressed:
1. Please provide the MIC values in the Abstract.
2. Line 110 – Should it be “oil-in-water” or the opposite?
3. Please provide the information about the solvent used in UV-Vis analysis.
4. Please create the Supplementary Data file showing all chromatograms obtained in the LC-ESI-MS analyses.
5. Line 424 - 250°C ???
6. As the comfrey root contains allantoin and other active substances, how can the Authors be sure that only polyphenolic acids are responsible for observed wound healing with their extract? What is the impact of the other substances?
Author Response
REV2
In the paper entitled "Healing of skin wounds in rats using creams based on Symphytum officinale extract", the Authors described the fabrication, characterization, and biological evaluation of polyphenolic comfrey root extract. The presented research is very insightful and worth publishing in IJMS. However, some minor issues have to be addressed:
- Please provide the MIC values in the Abstract.
A: Thanks for the observation. We are sure that now the abstract will be improved.
- Line 110 – Should it be “oil-in-water” or the opposite?
A: Yes, it is oil in water cream. The mistake was modified.
- Please provide the information about the solvent used in UV-Vis analysis.
A: The used solvent was water. This information was introduced in the manuscript.
- Please create the Supplementary Data file showing all chromatograms obtained in the LC-ESI-MS analyses.
A: Thanks a lot! According to your suggestion, we added a supplementary file with Figures S1 and S2.
- Line 424 - 250°C ???
A: Thank you very much! The correct is 25°C.
- As the comfrey root contains allantoin and other active substances, how can the Authors be sure that only polyphenolic acids are responsible for observed wound healing with their extract? What is the impact of the other substances?
A: Excellent question. Due to this question, we determined allantoin that is found in large quantities. You are right, polyphenolic compounds together with allantoin are responsible for the beneficial effects of the cream. We are sure, that with all the data, our manuscript is improved (see the revised manuscript and supplementary data).
Round 2
Reviewer 1 Report
Comments and Suggestions for Authors
The authors provided a revised version of the manuscript including aspects and responses to my inquiries. I consider the manuscript proper to be accepted in the journal.